# The Internal and External Factors That Determined Private Investment in Ecuador 2007–2020

Guido Macas-Acosta, Genesis Macas-Lituma and Arnaldo Vergara-Romero *

School of Economics, Universidad Ecotec, Km 13.5 vía a Samborondón, Samborondón 092301, Ecuador
* Correspondence: avergarar@ecotec.edu.ec

**Abstract:** This article studies how electoral processes and internal factors of the Ecuadorian economy affect the dynamics of the country's business expectations. The hypothesis that the free market and socialist political models in an economy generate different reactions in the expectations of the agents, according to the prevailing macroeconomic context, is tested. The empirical analysis is based on time series tools on quarterly data between 2006 and 2021. The results show that the dynamics of investment adjustment to the relationships of internal factors, electoral processes, and other variables explain 84% of this behavior. This is more accelerated in political contexts that promote the free market and maintain social, political, and economic stability, showing an overreaction of agents to negative economic news following the loss-aversion hypothesis.

**Keywords:** private investment; electoral processes; loss-aversion hypothesis; time series





## 1. Introduction

In the 1980s and 1990s, the temperature of the expectations of investors in Latin America constantly changed when they perceived mismanagement of fiscal and monetary policy, an exchange rate crisis, and the fragility of the financial system. This caused many economies in the region to strengthen their control institutions, make constitutional reforms, and even make changes in their monetary regimes to guarantee the sustainability of national and foreign private investment.

In the case of Ecuador, the most complicated decade was between 1996 and 2006, when presidential periods lasted less than two years and caused a lot of social, political, and economic instability. Financial shocks were difficult to control as they encouraged capital flight and devalued the currency. This storm ended with the fall of the Sucre, which until March 2000 ceased to be its currency to make way for the official dollarization of the Ecuadorian economy.

Likewise, the destabilizing factors of investment in Latin America originated from the different political visions of the rulers of each country, who could be aligned by the freedom of the market or by the gradual intervention of the State in the economy. Added to this is the growing role of social actors such as Indigenous people, trade unionists, the military, and even legislators in government decisions. This is especially true when macroeconomic policies affect the poorest, and their claims can end in some cases with the expulsion of rulers before the completion of their presidential term. On that list is Ecuador, Peru, Venezuela, and Argentina, among others.

Since 2005, most South American countries have been administered by liberal or socialist governments. Additionally, therefore, the behavior of national or foreign private investment has had different paths. The reports of the central banks of several countries indicate that the factor that slows down private investment is the political environment, because it does not generate a favorable environment or give confidence to entrepreneurs. In other words, political instability and the governance crisis affect the stability of companies and households in future consumption decisions and possible projects.

According to ECLAC (2021), between 2005 and 2009, South America received an average of 68,302 million USD in Foreign Direct Investment (FDI), of which Bolivia and Ecuador received 259 and 465 million USD, respectively. The countries that received the most resources were Brazil (32,331 million), Chile (12,170 million), Colombia (8890 million), and Peru (4978 million).

Analyzing the period from 2010 to 2019 and comparing the economies managed by socialist rulers, in that period, including countries such as Ecuador and Bolivia, the average FDI was located at 788.5 and 665.6 million USD, respectively. In contrast, Peru stood at an average of 8045 million with a mostly free market government.

Vergara, in his book "Citizens without a Republic", pointed out that in Peru, this was achieved because the money was invested in political, technical, and economic capital. "It did not fall from the sky. 17 trade agreements are not signed in 15 years by chance. The agenda of large private investment and the opening of our economy was decisively promoted" (Vergara 2013).

Ecuador and Bolivia, while governed by socialist models, never signed a trade agreement with either the United States or the European Union. Rather, they opted to strengthen themselves with countries of the same ideology, such as Cuba, Venezuela, China, Libya, Syria, Iran, and Russia. A special case is Chile, which maintained alternating socialist and liberal governments in that period, but the attraction of foreign capital rose to 17,840 million USD in the same period.

Without a doubt, investment is one of the most important components of any country, including private investment, which is one of the engines that drives the economy to increase employment levels and boost consumption. Thus, since 2003, many countries in the region have built indicators to measure the pulse of investors through expectation surveys. For example, in Peru, since 2003, there has been a three-month expectation index that is fed by a survey of macroeconomic expectations carried out by the Central Bank of the Republic of Peru. It serves to show the volatility and, with it, the uncertainty generated by political instability and the governance crisis (BCRP 2022).

In Ecuador, the Business Confidence Index (ICE) also began to be applied in 2007. It is an indicator carried out by the Central Bank of Ecuador and measures the perception of the business sector in terms of its economic activity in the national and international environment. According to the BCE, ICE aggregates four productive sectors of the country: Industry, Commerce, Services, and Construction. To achieve this, the information is collected through a monthly business opinion survey of the main executives of the 1000 largest companies in the country (according to their level of sales) from the four production sectors mentioned (BCE 2022).

In the same year, an analysis of expectations began to be carried out—the forecasts that economic agents make about the evolution of specific variables and their influence on the dynamics of the economy. This business confidence index became an instrument for analyzing its evolution in times of uncertainty which, together with other variables, allows its effects on investment to be analyzed.

In addition, the analysis of investment and its importance in the Ecuadorian economy has been widely studied. The academy and public policy makers worked on proposals for tax reforms, productivity, innovation, technology, and trade agreements, among other actors.

Keynes (1937) established that investment is the leading cause of economic fluctuations and compared it to "animal spirits": changes of optimism and pessimism that come from businessmen. Other studies by Harrod (1972)-Domar (1946) and Solow identified that investment is responsible for long-term growth. For this reason, it was pointed out that there is a relationship between expectations and investment, hence the importance of quantifying the evolution of business expectations and the effects on private investment.

The economic literature divides the study of expectations into adaptive and rational. In adaptive literature, individuals make predictions using past information on these variables. In other words, they carry past errors in their predictions. Generally, adaptive expectations assume that individuals make their predictions using past information, and in particular,

they learn from their mistakes. This means that expectations are corrected each period by a fraction of the discrepancy between the variable observed in the current and previous period. Therefore, the change in expectations occurs slowly, as they accumulate data from the past, also assuming that agents ignore new information in the future.

Therefore, this study contributes to the economic research of the country with an econometric model that demonstrates the relevance of expectations and their impact in terms of real private investment or private Gross Fixed Capital Formation in the country. This investment model allows us to identify the degree of sensitivity to expectations in different situations in the Ecuadorian economy, especially in terms of elections, social conflicts, ideologies, and other factors that affect the variable.

*Theoretical Framework*

The economic literature before the 21st century focused on studying how the country's macroeconomic environment (fiscal deficit, current account, capital account, and business cycle) and financial conditions (investors' expectations, areas of risk exposure of the banking system, and the degree of legal certainty). The objective was to determine the causes of the crises in Mexico, Asia, Brazil, Russia, Argentina, and Turkey.

Several theoretical models have been developed that try to explain the economic and financial crises in Latin America in the 1970s. Later, a model analyzed how countercyclical policies tried to stabilize investor mistrust in the region in the 1980s. Additionally, at the end of the 1990s, debates on the problems of company balance sheets and their impact on the investment capacity of the private sector were incorporated. In addition, themes of moral hazard and bank panics were included. For other authors, economic crises can only be explained by two types of investors: informed and uninformed.

Investment has been considered throughout history as one of the predominant factors of economic growth. Persky (2000) and Tobin (2005) have tried to understand and explain it, while others, such as Keynes (1937), have been interested in the behavior essential to human decisions, introducing expectations into the analysis.

Studies and analyzes of some institutional issues related to the crisis, such as the electoral cycle and the degree of government commitment, were also included. That is why the electoral processes in the 21st century are relevant to understanding the possible scenarios in investment expectations and even more so if the finalists have different ideological tendencies. On the one hand, there are institutions that promote public spending and generate an expulsion effect on private initiatives. Additionally, alliances with companies encourage the arrival of fresh resources.

In Argentina, Néstor Kirchner arrived in 2003, and then in 2007, his wife Cristina Fernández de Kirchner continued until 2015, when policy was changed to focus on reducing inequalities and improving the redistribution of wealth. Private investment stopped to make way for public investment. However, Trujillo points out that this project's incompletion generated uncertainty and political disputes that did not guarantee the leaders' continued power and gave way to a liberal president and defender of the market: Mauricio Macri (Trujillo 2019).

Macri bet on measures to liberalize the exchange rate and imports, as well as exemptions from taxes for sectors of the economy linked to agribusiness or the exploitation of natural resources (mega-mining). These focuses gave investors peace of mind, but that turn to the right did not last long, and he faced a social, political, and economic conflict that he could only govern until 2019, when he gave way to the left again in the form of Alberto Fernández. Additionally, these internal factors, electoral processes with two different tendencies, once again affected the expectations of investing.

The same also happened in Peru in 2006, when Alan García and Ollanta Humala, two candidates from different lines, faced each other in the second round. In this election, expectations and uncertainty were strong, but they were reduced when García won. However, in 2011, the nationalist Humala confronted the populist Keiko Fujimori again. The tension in the markets was evident, and even more so when Humala won, but expectations did

not improve. Additionally, in 2016, two pro-market candidates, Pedro Pablo Kuczynski and Keiko Fujimori, faced each other, and investment expectations improved after the electoral process.

The bubble of stability burst less than two years into the new leader's term; he resigned due to allegations of corruption. In the following two years, three interim presidents were counted, which undermined the confidence of businessmen. Additionally, in 2021, the leftist Pedro Castillo won, and with the victory, private investment began to move to other countries that promote free markets. A report from Comex Peru indicates that the expectation index of the Peruvian economy has been reflected throughout 2021 and has been located at values below 50 since April of last year.

In Ecuador, history repeats itself. Rafael Correa governed uninterruptedly from November 2006 until 2017 and then gave way to his ally Lenín Moreno. Correa clashed three times with candidates from the right in 2006, 2009, and 2013, which caused private investment expectations to collapse. The effect of the socialist Correa was felt in the 2009 and 2013 elections with more force. Institutional changes, limitations on imports, the creation of new taxes, seizures of companies, financial reforms, and an increase in public spending were the main measures that he applied in his period.

However, in 2017, his government took a turn as it bet on private investment and searched for allies such as the International Monetary Fund and the United States. Additionally, in 2020, the banker Guillermo Lasso beat Andrés Arauz, who was supported by Correa. The fact that the two candidates went to the second round altered the board of investors, who were worried about the socialist recipe again (see Figure 1).

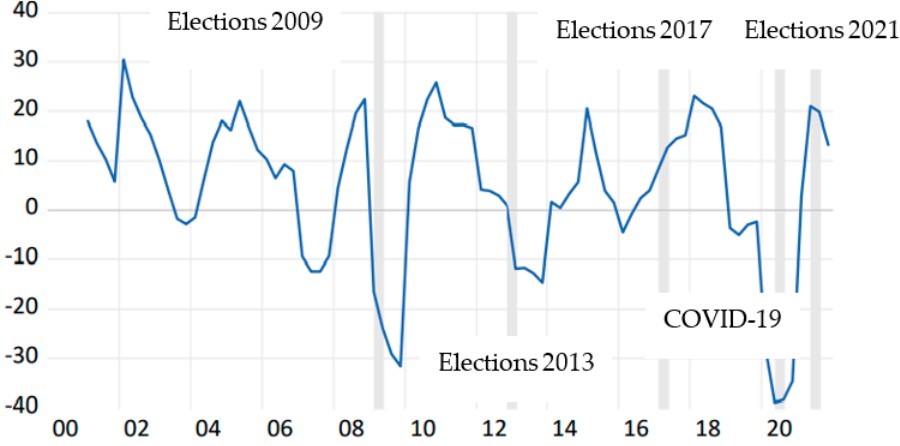

**Figure 1.** Behavior of private investment in Ecuador. Source: BCE (2022).

The presence of political instability has revealed a set of internal shocks, such as social, idiosyncratic, economic, and non-economic conflicts. Undoubtedly, investment expectations are marked by external factors, elections, and the country's internal factors.

In Latin America, there is little study material on expectations and their influence on investment. However, a study in Uruguay by Lanzilotta Mernies (2014) indicates that business expectations are procyclical to investment and anticipate investment by two quarters. In addition, it has been confirmed that in Uruguay, there is response asymmetry between favorable and unfavorable events, in which individuals are much more sensitive to negative scenarios than to positive events since market participants attribute a greater subjective value to losses than to earnings (see Figure 1).

The analysis of expectations or business confidence that economic agents have in the performance in the present and future of variables was conducted for the first time in a section of the General Theory by Keynes (1937). In this study, he concluded that the Great Depression was triggered by the New York Stock Exchange as a result of an abrupt drop in Aggregate Demand and not because of supply factors—that is, a fall in investor confidence about future profitability.

Studies of expectations begin with Keynes (1937), who points out that business investment decisions are due to waves of optimism and pessimism derived from uncertainty and confidence, respectively.

"Even setting aside the instability due to speculation, another instability that results from the characteristics of human nature: that much of our positive activities depend more on spontaneous optimism than on mathematical expectation, whether moral, hedonistic, or economic. Perhaps most of our decisions to do something positive, the full consequences of which will unfold for many days to come, can only be seen as the result of animal spirits—of a spontaneous spring to action rather than stillness, and not as a consequence of a weighted average of the quantitative benefits multiplied by the quantitative probabilities." (Keynes 1937, p. 144).

Binder and Kamdar (2022) analyze the role of expectations in depth, including how they intervene in inflation expectations and the effects of changes in their observed values. He showed that individuals form their expectations as a weighted sum of past expectations and inflation surprise, known as the Adaptive Expectations Theory.

Urbina Gárces (2017) carried out a similar study, concluding that business expectations directly determine Ecuadorian fixed investment and, in turn, exert an inelastic effect on fixed investment. Finally, this study showed that Ecuadorian investment is related to fundamental variables in the way that macroeconomic theory proposes.

Pinos Luzuriaga et al. (2019), in their research, "The role of private investment in the Ecuadorian economy", tried to measure the impact of fiscal variables on private investment and argued that public spending displaces private investment and that income tax distorts the decisions of economic agents against work, savings and investment. The authors used a panel data econometric model that includes Bolivia, Uruguay, and Venezuela.

On the other hand, Muth (1960) indicated the theory of expectations and concluded that economic agents match their predictions with reality based on a set of available information from the correct use of the theory. Further on, Lucas (1972) and Kydland and Prescott (1977) complement Muth's work and indicate that agents rationally formulate their expectations, using available information efficiently and minimizing the failure of their predictions. His study concludes that expansionary monetary policies only generate inflation.

These studies and models allow us to observe the effect caused by expectations on the economy of a country, which is why it is considered essential to generate and use expectation data. However, the inclusion of these expectations in economic models has some criticism. Kydland and Prescott (1977) points out that expectations and the microeconomic concept are unobservable, so establishing influence and measuring expectations' explanatory power is important.

## 2. Methodology

This investigative work proposes measuring the impact of political, social, and economic convulsions, ideologies, elections, and elections on private investment in Ecuador. This section explains the methodology used to verify the hypothesis that both electoral processes generate uncertainty in investment. An analysis was carried out with business expectations to verify if they were positively and inelastically related, using the period 2006-I–2021 IV. Using a time series database, a multiple regression model with dummy variables was used based on the ordinary least squares (OLS) model.

In order to break down the external factors, presidential elections, and internal factors that explain investment expectations and private investment, an econometric model was estimated in 2 stages:

1.  Model of investment expectations (Business Confidence Index) based on external variables and qualitative variables (Sökmen et al. 2021) on the 2017 presidential elections based on the following equation:

$$ICE = \beta_0 + \beta_1(TI) + \beta_2(PBI_{China}) + \beta_3(EMBIGA_{EC}) + \beta_4(D\_2009) + \beta_5(D\_2012) + \beta_6(D\_2017)$$
$$+\beta_7(D\_2020) + \beta_8(D\_2021) + \mu$$

where:

- ICE: Business Confidence Index expressed in points (Cruz et al. 2021; Huseynova et al. 2022).
- TI: Terms of Trade expressed in growth rates (Elhassan 2020; Vergara-Romero et al. 2022a).
- China's GDP is expressed in growth rates (Saygılı 2020).
- EMBIGA Ecuador is expressed in points (Cobham and Macmillan 2022; Olkiewicz 2022).
- D_2009: dummy variable for election period 2009Q1–2009Q2.
- D_2012: dummy for variable election period 2012Q4–2013Q1.
- D_2017: dummy for variable election period 2017Q1–2017Q2.
- D_2020: dummy for variable election period 2020Q2–2020Q3.
- D_2021: dummy for variable pandemic 2021Q1–2021Q2 (Alfaro 2005; Vergara-Romero et al. 2022b).
- μ: the stochastic disturbance term or random error.

2. Private investment model is based on external variables and qualitative variables on presidential elections, pandemics, and the residual (μ) of the model of the first stage, which approximates the component of investor expectations that is not explained by the external variables considered in the first stage (dynamics of idiosyncratic, economic, and non-economic factors).

$$\text{Priv\_inv} = \beta_0 + \hat{\beta}_1(\hat{\text{Exp}}) + \beta_2(\text{PBI}_{\text{China}}) + \beta_3(\text{Priv\_inv}_{t-1}) + \beta_4(\text{Publ\_inv}) + \beta_5(\text{EMBIGA}_{\text{Ec}}) + \beta_6(\text{D\_2009})$$
$$+ \beta_7(\text{D\_2012}) + \beta_8(\text{D\_2017}) + \beta_9(\text{D\_2020}) + \beta_{10}(\text{D\_2021}) + \mu$$

where:

- $\hat{\text{Exp}}$: ^ hat expectations as residues or factors that affect expectations (Bonilla 2017).
- TI: Terms of Trade, expressed in growth rates.
- Private investment has a lag (Tekin 2019).
- Public investment is expressed in growth rates (Chenet et al. 2021; Cobham and Macmillan 2022).
- EMBIGA Ecuador is expressed in points.
- D_2009, D_2012, D_2017, D_2020, and D_2021: dummy variables in electoral times and pandemics.

Annual and quarterly data were used for the period 2006Q1–2021Q4. Data on private investment, Business Confidence Index, and Terms of Trade come from the monthly technical report and national accounts prepared by the Central Bank of Ecuador. The EMBIGA information was used from the database of the Scope page.

In the first stage, the Business Confidence Index is used as an indicator that contains the expectations of four productive sectors: Construction, Industry, Commerce, and Services (Ochoa-Rico et al. 2022). Therefore, it is estimated that every month, this index gathers the opinion on the present and future of entrepreneurs on issues related to variations in sales, production, hiring of employees, inventory level, input prices, and perspectives of the business situation (Lux et al. 2020). This index meets the expectations of the Ecuadorian business sector.

With these data, the investment expectations model was developed based on external variables such as terms of trade, China's GDP, Ecuador's country risk, and election dummy variables. The strong commercial, financial, and economic relations and investments in hydroelectric projects that China has with Ecuador were one of the reasons for its selection as an independent variable.

For the second stage, private investment (FBKF) quantifies the acquisition and/or creation of fixed assets; these are responsible for most of the variation in total investment.

Before estimating the model of the two-time series stages, it was verified that the variables in quarterly periodicity do not present a non-stationary behavior, thus avoiding

ruling out spurious correlation problems. For this, the Newey–West correction estimator is applied, which is used to overcome autocorrelation, correlation, and heteroskedasticity in the model's error terms. In addition, the variables are expressed in growth rates and points (see Figure 2).

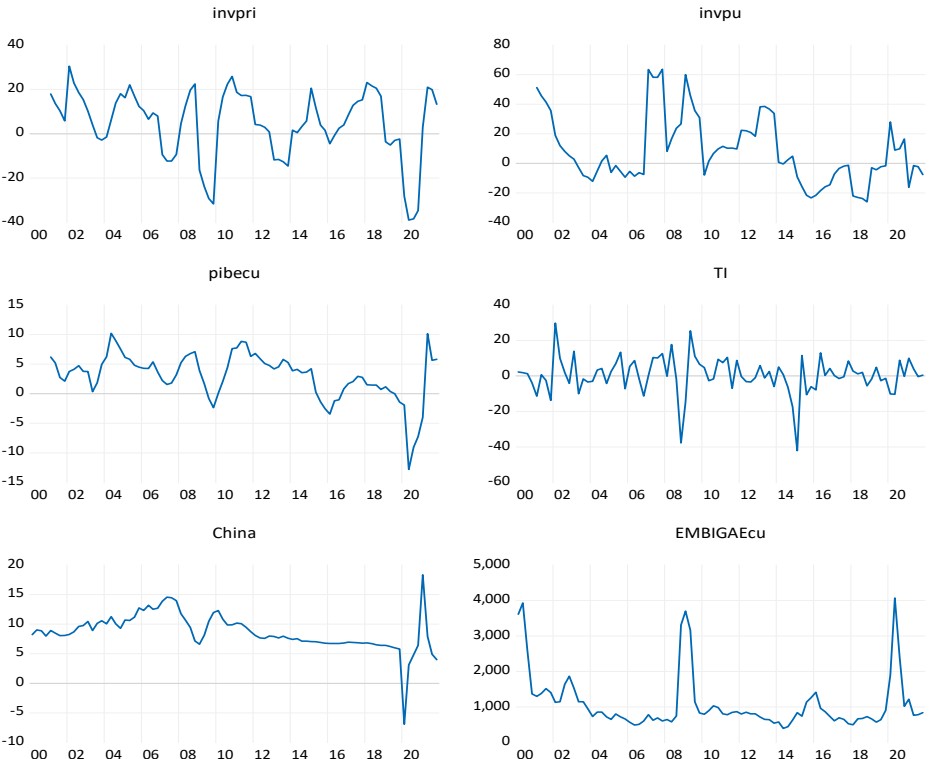

**Figure 2.** The behavior of the study variables. Source: BCE (2022).

## 3. Results

With the variables and the data, the first regression could be carried out. The purpose of this was to determine the unobservable factors that affect investment expectations and refer to other variables that are not included in the model. The short-term results of the model can be seen in Table 1; the variables considered are significant, with a probability value of the statistic (t-prob) less than 0.05. Among them are the terms of trade with a lag, China's economic growth, and country risk, but with a lag of seven.

The adjusted coefficient of determination ($R^2$) indicates that the external factors described above; the electoral processes of 2009, 2012, 2017, and 2021; and the 2020 pandemic influenced 39.3% of investors' expectations.

That means that the 61.7% difference is caused by other factors considered internal in the Ecuadorian economy, such as idiosyncratic, economic, and non-economic factors. Moreover, information captured in the model's residuals was included in regression two, as we tried to build an auxiliary regression.

These variables correspond to the internal factors that affect expectations and are called internal factors since they are non-observed data. This implies actions such as political instability, social conflicts, and idiosyncratic clashes, among others. Moreover, the graph of the residuals shows that there have been very pessimistic expectations in the election periods of 2009, 2017, and 2021. In 2020, there was a negative effect on expectations related to COVID-19, which affected macroeconomic variables worldwide.

In the first regression, internal factors explain 61% of investment expectations; the independent variables of the previous model explain 39% of the behavior of investment expectations. The residue is called hat expectations or internal factors, which are the unobserved factors that affect the certainty of investors.

**Table 1.** Econometric estimate of the expectation of private investment in Ecuador 2007–2021.

| | Estimate | Std. Error | t | *p*-Value |
|---|---|---|---|---|
| Var % TI (−1) | 0.2 | 0.070 | 2029 | 0.048 |
| Var. % GDP China | 1.1 | 0.470 | 2361 | 0.022 |
| Country risk Ecuador (−7) | 0.0 | 0.001 | 2521 | 0.015 |
| Elections 2009 | −6.0 | 2902 | −2076 | 0.043 |
| Elections 2012 | 4.6 | 1555 | 2945 | 0.005 |
| Elections 2017 | −8.6 | 2031 | −4250 | 0.000 |
| Elections 2021 | −12.3 | 4371 | −2818 | 0.007 |
| **Statistical Evaluation** | | | | |
| F-statistic | 5.62 | | *p*-value | 0.000 |
| R-squared | 47.9% | | Adjusted R-squared | 39.3% |
| **Residual Evaluation** | | | | |
| Normal | Jarque bera | 3486 | *p*-value | 0.17 |
| autocorrelation | Durbin-Watson | 1.73 | | |
| Heteroscedasticity | White | | *p*-value | 0.19 |

**Source:** Banco Central del Ecuador with Eviews.

The results of the variable elections in 2009 indicate that the election process—whereby President Rafael Correa, with a socialist ideology, was the leading actor—caused business confidence to drop by 6 points. Without a doubt, the new Constitution of the Republic prepared a path of control and regulation of the economy by the State.

In the 2013 elections, there were no adverse effects since President Correa won the elections again in a single round. In the electoral contest of 2017, the expectations of business people were again reduced by 8.6%; in 2021, they were the strongest, with a negative impact of 12%. At that time, socialist and free-market ideologies were popular again. In other words, investment expectations deteriorate in the election process.

Then, a second regression was carried out where the dependent variable was private investment. After evaluating the variables that contribute to the model, five determinants were selected: country risk, terms of trade, and hat expectations with their respective lags. To this, the dummy variable was created in each electoral process. Added to this is the dummy variable created in each electoral process used to measure the effects of expectations in each electoral contest. It was not unified into a single dummy variable because it did not generate separate results (see Table 2).

The variable representing internal factors has statistical significance and affects private investment. Raising unrealized expectations by 10 points improves investment by 4 percentage points after four quarters. In other words, if factors such as political instability or social conflicts are reduced, they can improve the arrival of resources to finance new investment projects in the next 12 months.

The electoral situation directly affects economic activity, reflecting lower confidence in the business sector. This translates to the paralysis of investment and hiring plans since investors do not have favorable information on the policies that the new administration will implement.

Another variable that affects private investment is public investment. When the state's gross fixed capital formation increases by 1 percentage point, it immediately translates into a 0.29 percentage point decrease in private investment. This means that an expulsion effect (crowding out) supports and defends the conventional theory.

It was also found that electoral processes have effects on private investment. In 2009, for example, if you face a presidential election where a radical candidate is running, it causes uncertainty in the market. This was the case with former President Rafael Correa, who, with his proposals to refund the state, caused a reduction in investment of 10 percentage points. According to information from the Central Bank of Ecuador (BCE), private investment was reduced by 2000 million USD in that year.

**Table 2.** Econometric estimate of private investment in Ecuador 2007–2021.

|  | Estimate | Std. Error | t | *p*-Value |
|---|---|---|---|---|
| Var % private investment (−1) | 0.84 | 0.08 | 6.45 | 0.00 |
| Var. %IT (−3) | 0.27 | 0.09 | 3.19 | 0.00 |
| Var. % GDP China (−2) | 1.16 | 0.51 | 2.26 | 0.03 |
| Country risk Ecuador (−7) | 0.00 | 0.00 | 2.89 | 0.01 |
| Public investment | −0.29 | 0.07 | −3.80 | 0.00 |
| **Internal factor (−4)** | 0.40 | 0.11 | 3.80 | **0.00** |
| 2009 elections | −10.42 | 4.60 | −2.26 | 0.03 |
| Elections 2017 | 7.11 | 2.27 | 3.13 | 0.00 |
| Elections 2021 | 21.69 | 2.85 | 7.61 | 0.00 |
| 2020 pandemic | −14.27 | 5.12 | −2.79 | 0.01 |
| Constant | −10.02 | 3.87 | −2.59 | 0.01 |
| **Statistical Evaluation** | | | | |
| F-statistic | 27.85 | | *p*-value | 0.00 |
| R-squared | 0.87 | | Adjusted R-squared | 0.84 |
| **Residual Evaluation** | | | | |
| Normal | Jarque Bera | 5.23 | *p*-value | 0.07 |
| autocorrelation | Durbin–Watson | 1.65 | | |

**Source:** Banco Central del Ecuador with Eviews.

At that time, the candidates for the Carondelet chair were the socialist Rafael Correa (AP), the former president Lucio Gutiérrez Borbúa (PSP), and the businessman Álvaro Noboa Pontón (PRIAN). In addition, the 22nd constitution had been drawn up in Ecuador, where the role of the state was key to the economy and was led by Correa, who ended up winning the election in a single round with 52% of the votes.

In the other electoral processes, there was no effect; the necessary capital was maintained to sustain its sales and market. Furthermore, in 2017, although the new candidate had the support of former President Correa, the political and economic management was different and allowed private investment to return to the market. When Lenin Moreno won, investment increased by 7% in the election period. Moreover, this is reflected even more strongly when banker Guillermo Lasso won the elections in 2021, and investment improved in growth rates by 22%. This confirms that investment expectations are closely related to private investment performance and more intensely in electoral processes.

## 4. Conclusions

Political uncertainty, social conflicts, idiosyncratic problems, and electoral episodes affect the perception of people in business and the future of investment plans in the economy. In addition, public investment and the electoral situation can directly affect economic activity through lower or higher confidence in the business sector. Furthermore, there is a reduction in investment plans, especially when the candidates with options to win the elections are very radical with their proposals. In 2009, 2017, and 2021, the expectations were negative since one of the candidates proposed socialist models that national or foreign people in business no longer liked.

In effect, investment and contracting plans are paralyzed by a lack of information on how favorable or not the next administration's policies will be, which in turn is reflected in lower economic growth. Moreover, in 2017, president-elected Lenín Moreno turned his government strategies around and allowed private investors to participate in economic recovery and multiply trade relations with his principal partners, such as the United States. Furthermore, the same thing happened in 2021 when banker Guillermo Lasso triumphed in the elections, and the international community gave him its support.

**Author Contributions:** Conceptualization, G.M.-A. and G.M.-L.; methodology, G.M.-A.; software, G.M.-L.; validation, G.M.-A., G.M.-L., and A.V.-R.; formal analysis, G.M.-L.; investigation, G.M.-A.; resources, A.V.-R.; data curation, G.M.-A.; writing—original draft preparation, A.V.-R.; writing—review and editing, A.V.-R.; visualization, G.M.-A.; supervision, G.M.-A.; project administration, A.V.-R. All authors have read and agreed to the published version of the manuscript.

**Funding:** This research received no external funding.

**Informed Consent Statement:** Not applicable.

**Data Availability Statement:** Not applicable.

**Conflicts of Interest:** The authors declare no conflict of interest.

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
