# Peer review of "The Internal and External Factors That Determined Private Investment in Ecuador 2007–2020"

_economies, doi:10.3390/economies10100248_

Round 1
Reviewer 1 Report
The abstract presents an attractive hypothesis to be tested, which could make a substantive contribution to the literature, especially for comparative studies in Latin America. However, this is not reflected in the introduction. I think it is necessary to strengthen the argument that the authors present in the introduction, as the form in which it is currently presented is more appropriate for a technical report and not for an academic document. The text would benefit from a richer contextualisation of the case being addressed, as well as indicating the distinctive elements that make the case attractive to readers from other parts of the world.
The theory section, meanwhile, corresponds to what one would expect from an academic text, as it includes both the canonical literature on the subject studied and more recent studies corresponding to the state of the art. At the same time, different topics from each are incorporated. However, it seems to me that the theory does not end up in a proper dialogue with what the authors propose. This is particularly evident in the presentation of Figure 1, which moves too quickly from more abstract theory to literature about concrete cases. One would expect the authors to discuss the tension between theory and their case more extensively in this section.
The methodology is adequate, both the method used and the way it is detailed. While I do not entirely agree with the way they use the independent variables, I think the authors could justify their inclusion by providing a brief discussion of their effect. For example, why is China's GDP used? Even though we all understand its importance, I think the authors should try to explain the causal relation (or not) between the expected behaviour of their regressors and the actual effect they have tested. However, I think what might become more controversial is concerning the authors' variable of interest: the confidence index. One would expect the authors to develop more of an argument about the index they are using. Why should one think that the index is correct? The paper needs to clarify the validity of the index and, ideally, compare it with other indices so that researchers in other countries can seek to replicate its methodology for their particular cases.
Finally, as I mentioned, the way the authors presented the problem and the results are more typical of a technical report. Right, the authors present the results, and we are clear about the conclusions that can be drawn from their statistical modelling, but from there the authors do not venture to explain the hypotheses put forward or even argue from the results. One would expect a paper published in a journal of this nature to have a lengthy discussion of the results, as well as include possible routes for researchers who want to replicate these methodologies for their countries or even go deeper into the same research presented by the authors.
Author Response
Dear reviewer Thanks for the advice that has improved our article and provides a substantial systematic and methodological improvement.
Best regards.

Reviewer 2 Report
The paper deals with an interesting topic, namely the different factors affecting the dynamics of Ecuador business expectations and the evolution of the country's Business Confidence Index.
However, although the research question is indeed very intriguing, the authors do not address it in a satisfying manner. The abstract, intro and literature review require extensive rewriting since there is need for language improvement but also so as to clarify how this paper compares to the relevant literature and how it extends it. More attention should be paid in mentioning related works on countries in Latin America but also on nations which are similar to Ecuador in other aspects, like political instability.
Regarding the empirical methodology, the sample is small for the number of variables considered and one suggestion could be to get rid of more dummies (maybe group elections?) or run multiple models, each with different exogenous variables and compare. The adjusted R squared is problematic in both models (too low for the Table 1 model and too high for the Table 2 one) and this issue is not properly addressed. Also the choice of variables is not properly justified or compared to the relevant literature. Finally, the results are not presented in a clear way, and neither is the paper's methodological innovation and how it extends the existing literature on investment.
I believe that the paper is not publishable as it is but, with a through rewriting and some methodological adjustments, it could become a valuable contribution to the field.
Minor comments:
1. Maybe the authors should leave out the Keynes quote (l. 96-102) because it is too big and weakens their main argument.
2. l. 21 Maybe write presidential terms instead of periods.
3. l. 39 Academia instead of academy.
4. l. 232 Why lag of 7? And why China in particular? Explain.
Author Response

(The authors gave the same response as above.)

Reviewer 3 Report
I find the paper generally interesting. I provide below some comments which could possibly contribute to its improvement:
1) The editing of the paper requires some major improvements.
a) For example, in the last paragraph of the Introduction, the author(s) state “Pinos Luzuriaga et al. (2018), in his research…” Since the authors are many (namely seven), it should be corrected to “in their research”. Moreover, in the References the year of publication is 2019.
b) Another example in page 3. The author(s) state that “Binder & Kamdar (2022) analyzes…”, instead of “Binder & Kamdar analyze...”.
c) In some references only the last names of authors are mentioned [e.g., Prescot (1977) and Lucas (1970)] while in some others and their first names [e.g., Urbina Gárces (2017)]. A uniform format should be followed. Moreover, in References the authors of the above paper are two, namely Urbina Gárces and Gabriel Alejandro.
d) Another example, in Methodology, first paragraph, the author(s) state two times the word elections (they state “elections and elections on private investment in Ecuador”.
2) In Figures 1 and 2 it should be clearly specified what is measured on the vertical axis.
3) The Methodology section requires major improvements.
a) The presentation of the statistical – empirical analysis should be more rigorous, explaining in more detail the selection of the dependent and explanatory variables and their form, as well as their weaknesses.
b) Some shortcuts should be explained (e.g., what is EMBIGA and why it is expressed in points) for those who are not familiar with Ecuador.
c) Why the author(s) specified four dummy variables for elections, that is D_2009, D_2012, D_2017 and D_2020 instead of using only one dummy for elections?
d) Why and how the author(s) selected the lagged form of variables?
e) Why the China’s GDP was introduced into the model as explanatory variable and not the effect of other countries?
f) More detailed analysis should be provided referring to the ICE since it is a central factor in the analysis.
4) The Results section of the paper requires more elaboration.
a) For example, by comparing Tables 1 and 2 we realize that the signs of some dummy variables differ between them. For example, Elections 2017 and Elections 2021 in Table 1 have negative signs while in Table 2 have positive signs. This outcome might imply instability of the model. The existence of instability should therefore be checked.
b) Some explanation should be provided as to why the signs of the election dummies differ.
c) In Table 2 the dummy Elections 2012 is not referred to. No explanation is provided for its omission from this model.
d) TI is the variable for the terms of trade, what is the IT variable in Table 2?
e) “2009 elections” in Table 2, should be renamed in “Elections 2009”.
f) Under what specific assumptions the Internal Factor (-4) was introduced into the Table 2 and why is it statistically acceptable?
Author Response
Dear reviewer Thanks for the advice that has improved our article and provides a substantial systematic and methodological improvement.Best regards.

Round 2
Reviewer 1 Report
Although I think the paper should elaborate on the conclusions presented, the authors have responded to the observations made. The paper can be published in its current form.
Reviewer 3 Report
The authors have made most of the requested revisions. I would therefore recommend the publication of the paper in its current form.